# Hypericin and Pheophorbide a Mediated Photodynamic Therapy Fighting MRSA Wound Infections: A Translational Study from In Vitro to In Vivo

**DOI:** 10.3390/pharmaceutics13091399

**Published:** 2021-09-03

**Authors:** Ben Chung Lap Chan, Priyanga Dharmaratne, Baiyan Wang, Kit Man Lau, Ching Ching Lee, David Wing Shing Cheung, Judy Yuet Wa Chan, Grace Gar Lee Yue, Clara Bik San Lau, Chun Kwok Wong, Kwok Pui Fung, Margaret Ip

**Affiliations:** 1Institute of Chinese Medicine and State Key Laboratory of Research on Bioactivities and Clinical Applications of Medicinal Plants, The Chinese University of Hong Kong, Shatin, N.T, Hong Kong 999077, China; benchan99@cuhk.edu.hk (B.C.L.C.); virginialau@cuhk.edu.hk (K.M.L.); cclee316@gmail.com (C.C.L.); dcwshk@gmail.com (D.W.S.C.); judychanyw@cuhk.edu.hk (J.Y.W.C.); graceyue@cuhk.edu.hk (G.G.L.Y.); claralau@cuhk.edu.hk (C.B.S.L.); ck-wong@cuhk.edu.hk (C.K.W.); kpfung@cuhk.edu.hk (K.P.F.); 2Department of Microbiology, Faculty of Medicine, The Chinese University of Hong Kong, Shatin, N.T, Hong Kong 999077, China; priyanga@cuhk.edu.hk; 3School of Biomedical Sciences, Faculty of Medicine, The Chinese University of Hong Kong, Shatin, N.T, Hong Kong 999077, China; tinawang@cuhk.edu.hk; 4Department of Chemical Pathology, Faculty of Medicine, The Chinese University of Hong Kong, Shatin, N.T, Hong Kong 999077, China

**Keywords:** photodynamic therapy, methicillin-resistant *Staphylococcus aureus*, hypericin, wound infection model

## Abstract

High prevalence rates of methicillin-resistant *Staphylococcus aureus* (MRSA) and lack of effective antibacterial treatments urge discovery of alternative therapeutic modalities. The advent of antibacterial photodynamic therapy (aPDT) is a promising alternative, composing rapid, nonselective cell destruction without generating resistance. We used a panel of clinically relevant MRSA to evaluate hypericin (Hy) and pheophobide a (Pa)-mediated PDT with clinically approved methylene blue (MB). We translated the promising in vitro anti-MRSA activity of selected compounds to a full-thick MRSA wound infection model in mice (in vivo) and the interaction of aPDT innate immune system (cytotoxicity towards neutrophils). Hy-PDT consistently displayed lower minimum bactericidal concentration (MBC) values (0.625–10 µM) against ATCC RN4220/pUL5054 and a whole panel of community-associated (CA)-MRSA compared to Pa or MB. Interestingly, Pa-PDT and Hy-PDT topical application demonstrated encouraging in vivo anti-MRSA activity (>1 log_10_ CFU reduction). Furthermore, histological analysis showed wound healing via re-epithelization was best in the Hy-PDT group. Importantly, the dark toxicity of Hy was significantly lower (*p* < 0.05) on neutrophils compared to Pa or MB. Overall, Hy-mediated PDT is a promising alternative to treat MRSA wound infections, and further rigorous mechanistic studies are warranted.

## 1. Introduction

Infections caused by antimicrobial resistant (AMR) bacteria are serious global health concerns and are exacerbated with prior asymptomatic carriage [1,2,3]. Methicillin-resistant *Staphylococcus aureus* (MRSA) is one of the commonest AMR bacteria that confers illnesses ranging from localized skin infections to systemic diseases, including toxic shock syndrome [4]. The prevalence of hospital-associated MRSA (HA-MRSA) infection varies geographically, and Hong Kong is one of the high-prevalence regions in Asia. According to the Asian Network for Surveillance of Resistant Pathogens (ANSORP) study, 57% of all inpatient isolates of *S. aureus* from Hong Kong hospitals were confirmed as methicillin-resistant [5]. Additionally, in Hong Kong, the prevalence of nasal carriage of *S. aureus* and MRSA were 27.6% and 1.3%, respectively, among children in daycare centers and kindergartens [6,7].

Microbes by their nature continually adapt to survive the antimicrobial treatments we use to combat them, resulting in an ever increasing level of antimicrobial resistance [8], and the development of nonantimicrobial treatments may be beneficial concerning resistance development. Photodynamic therapy (PDT) consists of the administration of a nontoxic drug or dye known as a photosensitizer (PS) either systemically, locally, or topically applied to a patient, followed by illumination with visible or near-infrared (NIR) light in the presence of oxygen, leading to the generation of cytotoxic reactive oxygen species (ROS) in the proximate environment causing cell death/ tissue damage [9,10]. The advantages of PDT over conventional therapies include rapid bacterial killing, applicability over a broad spectrum (Gram-negative or Gram-positive) [11,12], and efficacy against biofilms [13,14], fungi [15,16], parasites [17] and viruses [18]. To date, the clinical applications of PDT have been confined mainly to localized infections in dermatology and dentistry [19,20], wound healing [21,22], and for surface disinfection including medical devices [23].

It was reported that PDT for localized microbial infections exerts its therapeutic effect both by direct bacterial killing and the activation of the host immune response, particularly innate immunity [24]. Neutrophils are among the first line of defence recruited to the site of infection to release enzymes for killing infectious organisms and to secrete cytokines that promote inflammation. The importance of neutrophils against microbial infections is reflected by the observation that Photofrin ^®^-PDT exhibited significant cytotoxicity for cultured MRSA, but the therapy had a low efficacy in a murine model of MRSA arthritis, even though Photofrin^®^ accumulated well in the infected joint. It was discovered that 30% of intra-articular leukocytes, mainly neutrophils, were killed immediately during or following Photofrin-PDT [25]. Therefore, we assume it is important to examine specific PS-PDTs cytotoxicity towards human neutrophils.

Hypericin (Hy) is a naturally occurring polycyclic quinine (Figure 1a) extracted from plant species of the genus *Hypericium* including the species *Hypericum perforatum* L. (St John’s Wort) [26]. Recent reports showed that Hy has the potential to treat several types of cancer and some benign skin disorders [27,28]. Interestingly, Yow et al. reported Hy could induce a significant cytotoxic effect on clinically isolated methicillin-sensitive *S. aureus* (MSSA) and MRSA [29]. In the aspect of wound healing, *H. perforatum*, which is a popular folk remedy for the treatment of wounds in Turkey, has been shown to possess remarkable in vivo wound healing activity, and Hy was found in the active fractions [30].

Pheobhobide a (Pa) is also a natural compound, derived from the breakdown of chlorophyll a [31]. The extended π-π conjugated system (Figure 1b) and stability of the compound in various solvents make it suitable as a photosensitizing agent. Studies have revealed that Pa-PDT is effective in eradicating a variety of tumors, including pigmented melanoma, colonic cancer, Jurkat leukemia, and pancreatic carcinoma [32,33,34,35]. Besides the anticancer activity of Pa, it has also been tested for its photodynamic activity against MRSA with modification to its structure (Na salt of Pa) [36].

Photophysical properties are of paramount importance when selecting a photosensitizer. The absorption spectrum of the compound plays a pivotal role during in vivo applications, and it has to be within the therapeutic window (550–950 nm) [37]. The key photophysical properties of Hy and Pa are summarized in Table 1 along with the gold standard of PDT studies (Methylene blue, MB, Figure 1c).

The compelling evidence of Hy and Pa led us to investigate their PDT effects in vitro and in vivo against a broad spectrum of clinically relevant MRSA panels along with their toxicity towards neutrophils, in view of depicting their overall anti-MRSA efficacy.

## 2. Materials and Methods

### 2.1. General

Pheophorbide a was purchased from Frontier Scientific Inc. (Logan, UT, USA) and hypericin and methylene blue were purchased from Sigma-Aldrich Co. (St Louis, MA, USA). The PS solution for in vitro PDT study was prepared freshly by dissolving Pa and Hy in DMSO to make a 10 mM stock solution. It was then diluted in Tween 80 and MHB to set the desired stock solution. A serial two-fold dilution procedure was employed to obtain final working concentrations. Tween 80 and DMSO concentrations were maintained ≤ 0.1% and ≤1% (*v/v*), respectively, in each test group.

The bacterial strains MRSA, ATCC 43300, ATCC BAA-42, ATCC BAA-43, ATCC BAA-44, two mutant strains [AAC(6)′ APH(2)′′ and RN4220/pUL5054], five community-acquired (CA-MRSA) and five hospital-acquired MRSA (HA-MRSA) clinical strains were obtained from the Department of Microbiology, Faculty of Medicine, The Chinese University of Hong Kong.

### 2.2. In Vitro Photodynamic Minimal Bactericidal Concentration (PD-MBC) Studies

Minimal bactericidal concentrations (MBCs) of Pa-PDT, Hy-PDT and MB-PDT for sixteen MRSA strains were determined according to the modified method adopted by Clinical and Laboratory Standards Institute (CLSI) guidelines [12,44,45]. Briefly, an overnight bacterial culture suspension was adjusted to McFarland Standard 0.5 and suspended in Mueller Hinton Broth (MHB) to make a final concentration of 1.0×10^6^ colony forming unit (CFU)/mL. Photosensitizers at different concentrations (100 µL) and MRSA suspension (100 µL) were added into 96-well plate and incubated at 37°C for 2 h under dark condition as a pre-irradiation step. After incubation, the mixed solutions were irradiated from above at a light intensity of 40 mW/cm^2^ using a 300 W quartz-halogen lamp attenuated by a 5 cm layer of water as a heat buffer and a color filter cut-on at 610 nm (for MB and Pa, λ ≥ 610 nm) or 590 nm (for Hy, λ ≥ 590 nm) for 20 min, i.e., 48 J/cm^2^. Dark control group and a solvent control group were included. All experiments were repeated three times. The MBCs were determined as the minimum concentration of the photosensitizers required for complete inhibition of bacterial growth on a blood agar plate.

### 2.3. Animal Studies-Mouse Model of MRSA-Infected Wound

#### 2.3.1. Animal Model

Previously published murine skin infection models [44,45,46,47] were used to validate the in vivo efficacy of the PS-PDT treatment against MRSA. All animal experiments were conformed to the university guidelines and approved by the Animal Experimentation Ethics Committee (Ref. no.12/076/MIS, 8 February 2013) of The Chinese University of Hong Kong. Female Balb/c mice (25–30 g) were supplied by Laboratory Animal Services Centre (LASEC), The Chinese University of Hong Kong. They were housed in individually ventilated cages (IVC) under the conditions of 22–25 °C and a 12-h light-dark cycle, with free access to chow and tap water.

It is apparent from the in vitro results that, MRSA ATCC RN4220/pUL5054 strain was susceptible for all three PSs with comparatively lower MBC values. Hence, this selected to establish infection on a full-thick wound in mice. Mice were anesthetized by an intraperitoneal (i.p.) injection of ketamine (40 mg/kg) and xylazine (8 mg/kg), with the hair of the back shaved and the skin cleansed with 10% povidone-iodine solution. A circular full-thickness wound (4 mm in diameter) was established through a disposable skin puncher on the back subcutaneous tissue of each animal. The lesion, overlaid with gauze, was dressed with an adhesive bandage.

For the in vivo studies, Pa and Hy were prepared according to our previously published protocol [48]. Briefly, 1% DMSO, 4% ethanol and 95% PBS constituted the final test solution.

#### 2.3.2. Intravenous Treatment

Our research group previously investigated Pa-PDT-mediated anticancer activity (against MCF-7 tumors) in vivo [48]. So, the dosage and optimum therapeutic window for these kinds of compounds were established (2.5 mg/Kg) for intravenous injection. Three days after wound induction, mice were anesthetized with a ketamine/ xylazine cocktail, and 50 µL of MRSA (1 × 10^8^ CFU/ mL) was inoculated onto the wound. One day later, 20 µL of photosensitizers (Pa or Hy) at 2.5 mg/kg were intravenously injected into the mice via the tail vein as for the stratified groups. Ten minutes after the application of photosensitizers, PDT illumination at 1 W was performed for either 30 s or 10 min for all PDT groups, corresponding to 30 J/wound and 600 J/wound, respectively. A continuous-wave laser was generated from the Ceralas medical laser system with excitation at 670 nm (Biolitec group, Bonn, Germany). The treatments were repeated every other day and lasted until Day 8 (three treatment cycles, Figure 2). Wound sizes were recorded before and after treatment. At the end of the experiment, animals were euthanized with an overdose of terminal pentobarbital solution. The wound (5 × 10 mm) was then excised aseptically.

#### 2.3.3. Topical Treatment

Three days after wound induction, mice were anesthetized with a ketamine/ xylazine cocktail and the adhesive bandage was removed. A 50 µL of MRSA suspension (1 × 10^8^ CFU/ mL) was dropped onto each wound. A dressing (Tegaderm^TM^ film, 3M, Company, St. Louis, MA, USA) was applied to cover the wound immediately to maintain wound moisture. Thirty minutes after bacterial inoculation, 50 µL of 800 µM PS solutions or Fucidin^®^ cream was injected under the dressing by syringe and allowed to spread over the wound. Photoactivation (Biolitec group, Bonn, Germany) was initiated immediately. A single dosage of laser at 0.5 W for 60 s was delivered by an optical fiber 2 mm in diameter, corresponding to 30 J/wound. The dark control (PS alone) groups and the Fucidin^®^ cream (2% fucidic acid) group (positive control) did not receive any laser irradiation but were sham-irradiated under visible light. The animals were returned to individually ventilated cages (IVC) after treatment and thoroughly examined daily. To avoid any possible phototoxicity, all mice were kept in a dark room for 4 h after PDT/sham irradiation. After 2 days of treatment, once daily (Figure 3), the dressings were removed and the wounds were exposed. The wound sizes were recorded every two days. On Day 10, animals were euthanized with an overdose of dorminal pentobarbital solution. The wound (5 × 10 mm) was then excised aseptically.

Each skin sample was divided into two portions. One-piece was used for histological examination to determine the maturity of wound repair, and the second was weighed and homogenized in 0.5 mL of PBS solution for bacterial viability counts. Quantification of viable bacteria was performed by culturing serial dilutions (10 μL) of the bacterial suspension on blood agar plates. For this purpose, all plates were incubated at 37 ^0^C for 24 h and evaluated for the presence of the staphylococcal strain. The bacteria were quantified by counting the number of CFU per plate.

#### 2.3.4. Histological Evaluation

Wound tissues collected from the animal study were initially fixed in 10% buffered formalin, followed by dehydration and paraffin-embedding. Paraffin blocks were cut into 5 μm tissue sections including the epidermis, the dermis, and the subcutaneous panniculus. The sections were stained with hematoxylin and eosin (H&E) and assessed by light microscopy for wound healing.

### 2.4. In Vitro Cytotoxic MTT (3-(4,5-Dimethylthiazol-2-yl)-2,5-diphenyltetrazolium Bromide) Assay on Human Neutrophils

Human neutrophils were purified from the fresh buffy coat fraction of blood from adult volunteers at the Hong Kong Red Cross Blood Transfusion Service, Hong Kong and separated by the Percoll method which was routinely performed in our laboratory [49].

Our studies showed that human neutrophils exhibited a short lifespan after isolating from buffy coats. Most human neutrophils did not survive 48 h after isolation. Therefore, freshly isolated human neutrophils were used in the present study, and the experiments were done within 24 h after isolation. Freshly isolated human neutrophils were plated in 96-well plates at 10^5^ cells/well. Serial dilutions of three photosensitizers, Pa, MB, and Hy were added to the wells. After 24 h at 37 °C incubation, MTT solution (50 μL, 5 mg/ mL) were added to each well. Then, the plates were incubated at 37 °C for 3 h. After incubation, 150 µL of DMSO was added to each well. The OD of the wells was determined by a spectrophotometer at 590 nm. Toxicity was represented by the ratio of OD of a well in the presence of compounds with the OD of control wells in the presence of a medium containing DMSO.

## 3. Results

### 3.1. Bactericidal Activity Assay on MRSA Strains

It is apparent from Table 1 that the Pa-PDT group showed significantly higher (*p* < 0.05) anti-MRSA activity against MRSA ATCC RN4220/pUL5054, W44, and W46-47 (MBC; 3.125–12.5 µM) than the positive control MB (MBC; 120->160 µM). Similarly, the Hy-PDT group demonstrated significantly higher (*p* < 0.05) anti-MRSA activity against MRSA ATCC RN4220/pUL5054 and a whole panel of CA-MRSA strains (MBC; 0.625–10 µM) compared to MB. However, HA-MRSA was more resistant towards HY-PDT, except HA-232 (MBC; 2.5 µM). Interestingly, Hy-PDT showed the lowest MBC values compared to Pa-PDT or MB-PDT, indicating the importance of further investigations. The dark toxicities of all three PSs were 4-8 times lower than their PDT counterparts (Table 2). Out of these sixteen MRSA strains tested, RN4220/pUL5054 was sensitive to three photosensitizers, especially to Hy and Pa. Therefore, it was selected to establish the in vivo model.

### 3.2. Animal Studies-Mouse Model of MRSA-Infected Wound

#### 3.2.1. Effect of PDT of Pa and Hy with Intravenous Injection in MRSA-Infected Wound Model

Neither of the PDTs (30 s or 10 min) upon Pa and Hy intravenous (i.v.) injection (2.5 mg/kg) significantly (*p* < 0.05) reduced bacterial load at Day 8 (Figure 4). It was observed that mice receiving 30 s of Pa-PDT and Hy-PDT treatments resulted in slight but insignificant promotion of wound closure. However, this trend could not be observed in the 10 min PDT-treated groups (Figure 5). There was no body weight loss in treatment groups when compared with the control group, implying that the treatments did not cause distress in the mice (Figure 6).

#### 3.2.2. Effect of PDT of Topically Applied MB, Pa or Hy in MRSA-Infected Wound Model

Topical application of Fucidin cream eradicated MRSA in the wound. Pa-PDT and Hy-PDT treatment groups showed significant antibacterial effects against MRSA when compared with the no treatment group (1 log decrease of CFU, *p* < 0.05) (Figure 7). The size of Fucidin cream-treated wounds was slightly larger and there was no great difference in wound sizes among all other groups after treatment (Figure 8). There was no body weight loss in the treatment groups when compared with the control group (Figure 9), implying that the treatments did not cause distress in the mice.

#### 3.2.3. Histological Evaluation

Histopathological assessment of untreated wounds on day 8 (Figure 10) indicated incomplete epithelialization, loose granulation tissue with areas of poorly stained extracellular matrix where collagen fibers were either immature or lacking as a sign of obvious ulcer formation. It is apparent from Figure 10, Hy and Hy-PDT mediated groups showed rather good wound healing compared to the treatment naïve group by showing epithelial cells and fibrous tissue proliferation. MB-PDT had a minor healing effect in granulation and collagen formation and its re-epithelialization was worse than that of the no treatment group. Wound healing of the Fucidin cream-treated group was worse than the no treatment group (Figure 10).

### 3.3. Cytotoxicity Effect of Pa-PDT, Hy-PDT or MB-PDT on Human Neutrophils

Three photosensitizers, Pa, MB and Hy, were incubated with human neutrophils for 24 h. No light irradiation was applied to the photosensitizers. Viability of human neutrophils was determined by MTT assay. As shown in Figure 11, Pa and MB were more cytotoxic to human neutrophils at 24-h incubation with LC_50_ at ~10.16 µM and ~11.22 µM, respectively, whereas Hy showed a LC_50_ higher than 50 µM.

Cytotoxicity of the three photosensitizers with light irradiation for 20 min (48 J/cm^2^) on human neutrophils was also examined. As shown in Figure 12, Hy-PDT possessed the strongest cytotoxicity with LC_50_ less than 3 µM, whereas LC_50_ of Pa-PDT and MB-PDT were ~4.44 µM and ~4.23 µM, respectively. As predicted, drugs with light irradiation exhibited significantly higher cytotoxicity than drugs without light irradiation. Given the high cytotoxic properties of photosensitizers with light irradiation, we expected that no cytokines would be produced in human neutrophils, as well as in the photosensitizer-administered wound sites when light irradiation was applied.

## 4. Discussion

PDT with MB is widely tested to be effective against Gram-positive bacteria, including MRSA, and acts as the gold standard for efficacy comparison for PDTs with Pa and Hy. In vitro results showed that Hy-PDT and Pa-PDT against some MRSA strains had better bactericidal activity than MB-PDT. Among the sixteen MRSA strains tested, the lowest MBC for PDTs with Hy, Pa, and MB were 0.625 µM (0.5 µg/mL), 3.125 µM (2 µg/mL), and 80 µM (32 µg/mL) respectively. Therefore, Hy-PDT and Pa-PDT exhibited potent bactericidal activity on MRSA strains. Hy possesses appropriate photochemical and photobiological properties, such as a high singlet oxygen quantum yield and cytoplasmic membrane localization, which makes it suitable for use as a PS in PDT [29,50]. Furthermore, its absorption maxima (λmax) 570 nm at longer wavelength [51] makes Hy suitable for PDT because of its high penetration ability. The obtained in vitro results for Hy are comparable with the previously published data in Yow et al. [29] for two CA-MRSA strains (W 45 and W 47, Table 2) where a > 6 log_10_ CFU reduction of MRSA was obtained at an 8 µM concentration and 30 Jcm^−2^ light dose. However, MBC values for RN4220/pUL5054, W44, W46 and W48 showed significantly lower (*p* < 0.05) MBC values compared to the published report.

In addition, it was found that Pa-PDT inhibited P-glycoprotein-mediated multidrug resistance via c-Jun N-terminal kinase (JNK) activation in human hepatocellular carcinoma [52]. As P-glycoprotein is an important class of efflux pumps that is always associated with a high prevalence of antibiotic resistance, it is hypothesized that Pa can circumvent drug resistance in MRSA as well.

To evaluate the efficacies of photodynamic therapy mediated by Pa and Hy, an MRSA-infected wound-bearing mice model was used. We found that MRSA was far more resistant to PDT in the more complicated environment of the murine dorsal wound than in transparent Petri dishes in vitro. It is possible because the wound tissue provided more layers of organic material to scatter light and to host the bacteria. The open wound of skin tissue, as the natural colony of Staphylococci, might provide a more nutritious matrix for bacterial survival and prosperity, and the aqua dependency of the cytotoxicity of photosensitizers might also partly be impeded in the relatively lower moistures microenvironment of the wounds.

However, the bodyweight of all mice was weighed before and after 8 days of treatment and there was no difference between groups and there was also no significant behavioral change in the mice, indicating that wound induction, MRSA infection, and our treatments had little effect on the general health status of the mice.

It was observed that intravenous injection of photosensitizers with PDT treatments had no antibacterial effect. This may be because none of the photosensitizers had an affinity to wound tissue, so the circulating photosensitizers had little opportunity to aggregate at the wound to generate an adequate amount of ROS by light illumination to kill MRSA. We found that 10 min of PDT resulted in burnt scab formation shortly after light illumination in some cases, while 30 s of PDT did not. The 30 s of PDT treatment also resulted in better wound healing.

The in vivo antibacterial effect of Hy-PDT and Pa-PDT was found to be significant and also stronger than MB-PDT at the same concentration, suggesting encouraging antibacterial effects of them against MRSA wound infection. However, it is interesting that the complete elimination of MRSA by Fucidin cream treatment was accompanied by worse wound healing as reflected by large open wound areas. It was also surprising that the histological appearance of wound healing in MRSA-infected wounds receiving only water and light illumination was significantly higher than that of No treatment group. We also observed more abundant vessel formation in the dermis of some samples than in any other group without significant improvement in open wound area. It has been reported that red light illumination promoted wound healing by promoting ATP release from mitochondria, activating the lymphatic system, increasing blood circulation and forming new capillaries [53]. Since we did not do further cellular nor molecular analyses, we are not in a position to explain the exact reason behind the varied wound closures among different treatment groups.

We tested the wound healing effects of several commercially available antibacterial-agent-free moisturizer skin creams and none of them promoted nor inhibited skin wound healing (data not shown). Pa-PDT, MB-PDT, and Hy-PDT were all toxic to neutrophils in vitro but the neutrophils were more resistant to Pa, MB, and Hy treatments without light illumination. Hy and MB were less toxic, while Pa showed some toxicity even without light illumination. Since our treatments did not include whole body illumination, the topically applied photosensitizers and locally irradiated light illumination could hardly have any toxic effects on the immune system inside the body. In addition, insignificant body weight differences among all groups also indicated the minimum effect of our treatments on the general health of animals.

## 5. Conclusions

Hy-PDT, Pa-PDT and MB-PDT were all capable of killing MRSA in vitro, and Hy-PDT showed highest efficacy against panel of MRSAs. Both Hy-PDT and Pa-PDT showed better efficacy than MB-PDT in antibacterial and wound healing effects against MRSA-infected wounds in our murine model, and the efficacy of Hy-PDT was the best among all treatments tested.

## Figures and Tables

**Figure 1 pharmaceutics-13-01399-f001:**
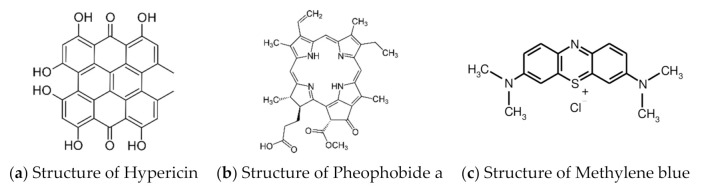
Chemical structures of 3 PSs used in the current investigation.

**Figure 2 pharmaceutics-13-01399-f002:**
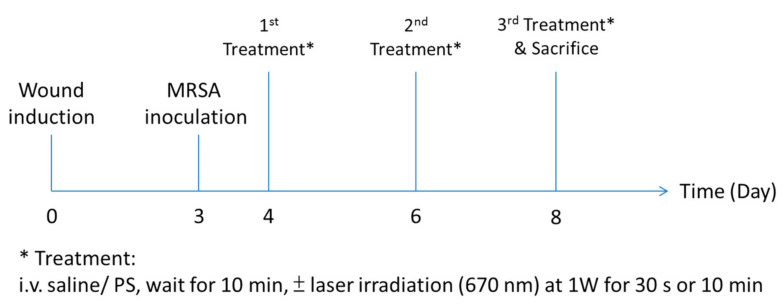
Timeline for the intravenous treatment.

**Figure 3 pharmaceutics-13-01399-f003:**
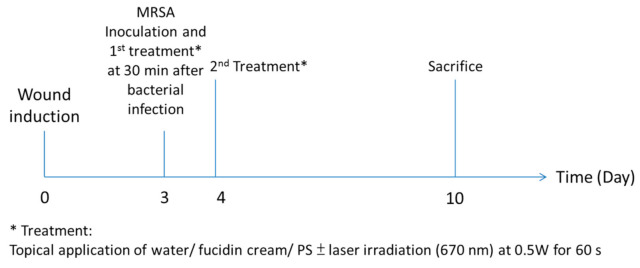
Timeline for the topical treatment.

**Figure 4 pharmaceutics-13-01399-f004:**
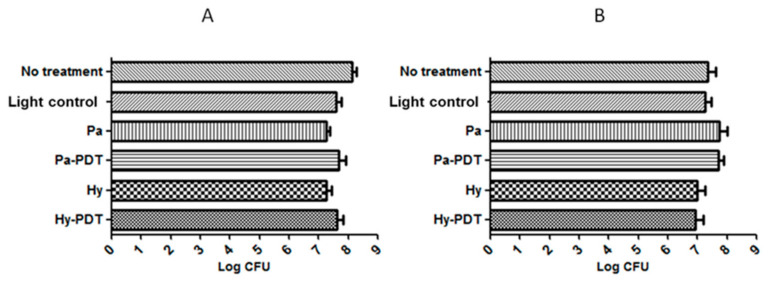
Bacterial load of wounds in different groups of mice after intravenous injection treatment with or without PDT. (**A**) Pa and Hy 2.5 mg/kg, irradiation for 30 s for PDT groups; (**B**) Pa and Hy 2.5 mg/kg, irradiation for 10 min for PDT groups. Data are mean ± SEM (*n* = 5).

**Figure 5 pharmaceutics-13-01399-f005:**
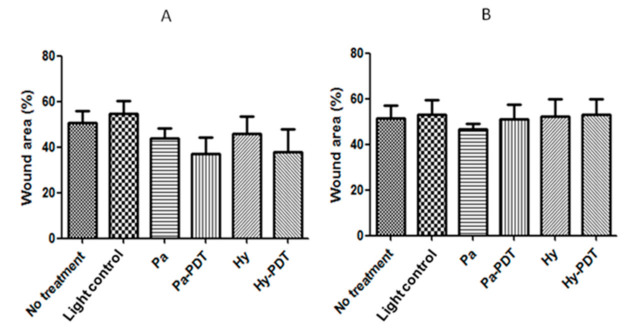
Wound areas in different groups of mice after intravenous injection treatment with or without PDT. (**A**) Pa and Hy 2.5 mg/kg irradiation for 30 s for PDT groups; (**B**) Pa and Hy 2.5 mg/kg, irradiation for 10 min for PDT groups. Data are mean ± SEM (*n* = 5).

**Figure 6 pharmaceutics-13-01399-f006:**
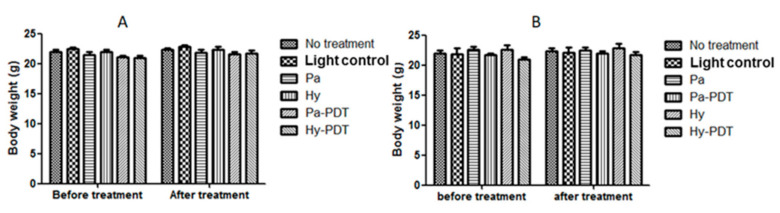
Body weight of different groups of mice after intravenous injection treatment with or without PDT. (**A**) Pa and Hy 2.5 mg/kg, irradiation for30 s for PDT groups; (**B**) Pa and Hy 2.5 mg/kg, irradiation for 10 min for PDT groups. Data are mean ± SEM (*n*=5).

**Figure 7 pharmaceutics-13-01399-f007:**
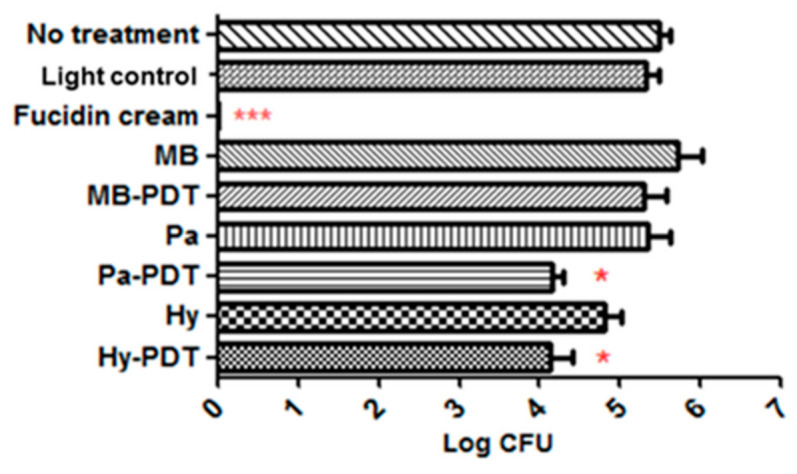
Bacterial load of wounds in different groups of mice after topical treatment with or without PDT. Data are mean ± SEM (*n* = 6–10).* *p* < 0.05 and *** *p* < 0.001 indicated significant bacterial load difference between No treatment and treatment groups by Student’s *t*-test.

**Figure 8 pharmaceutics-13-01399-f008:**
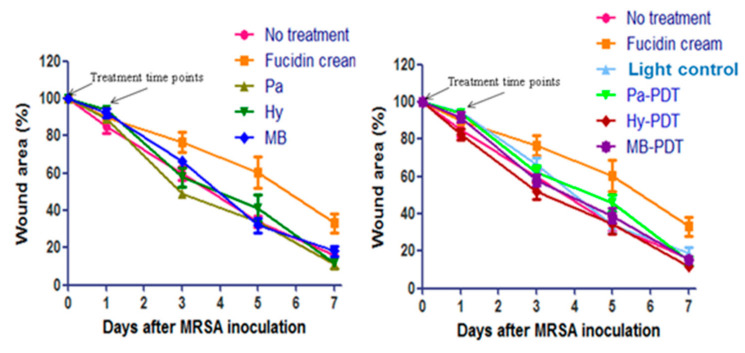
Wound areas of mice in different groups of mice after topical treatment with or without PDT. Data are mean ± SEM (*n* = 6–10).

**Figure 9 pharmaceutics-13-01399-f009:**
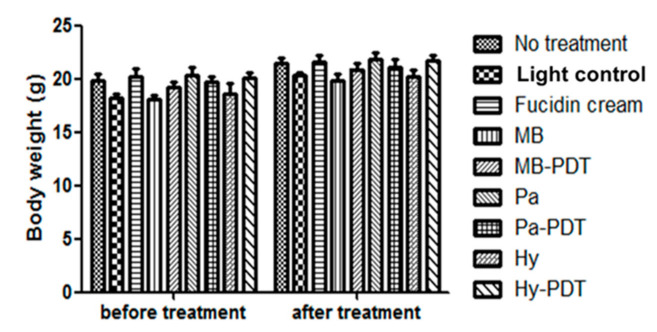
Body weight of different groups of mice before and after treatment with or without PDT. Data are mean ± SEM (*n* = 6–10).

**Figure 10 pharmaceutics-13-01399-f010:**
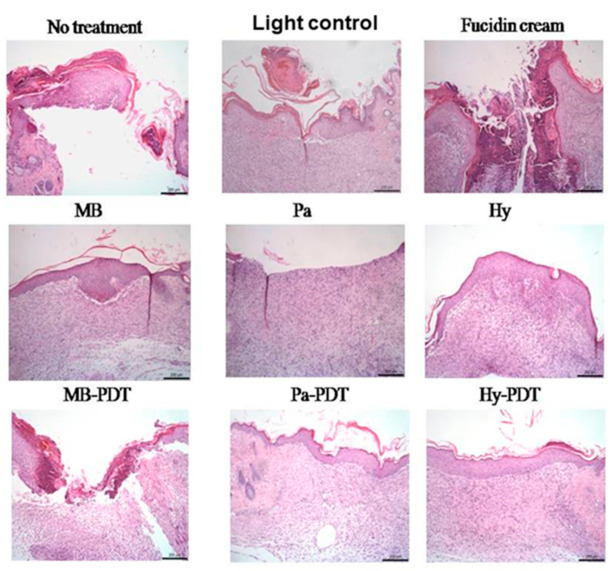
Representative wound section with haematoxylin and eosin stained and examined at ×100 magnification. Scale bar, 200 μm.

**Figure 11 pharmaceutics-13-01399-f011:**
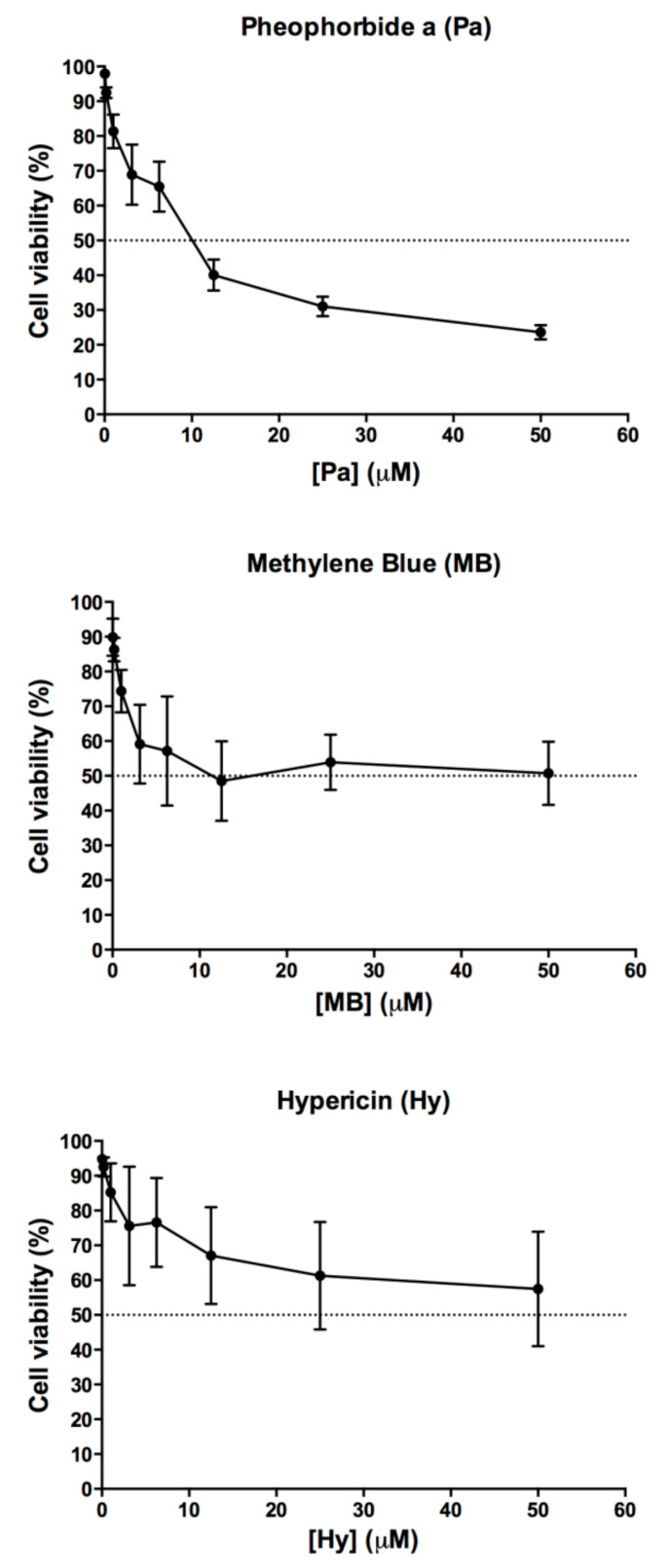
Cytotoxicity of Pa, MB and Hy, without light irradiation on human neutrophils isolated from buffy coat. Data are mean ± SEM (*n* = 3).

**Figure 12 pharmaceutics-13-01399-f012:**
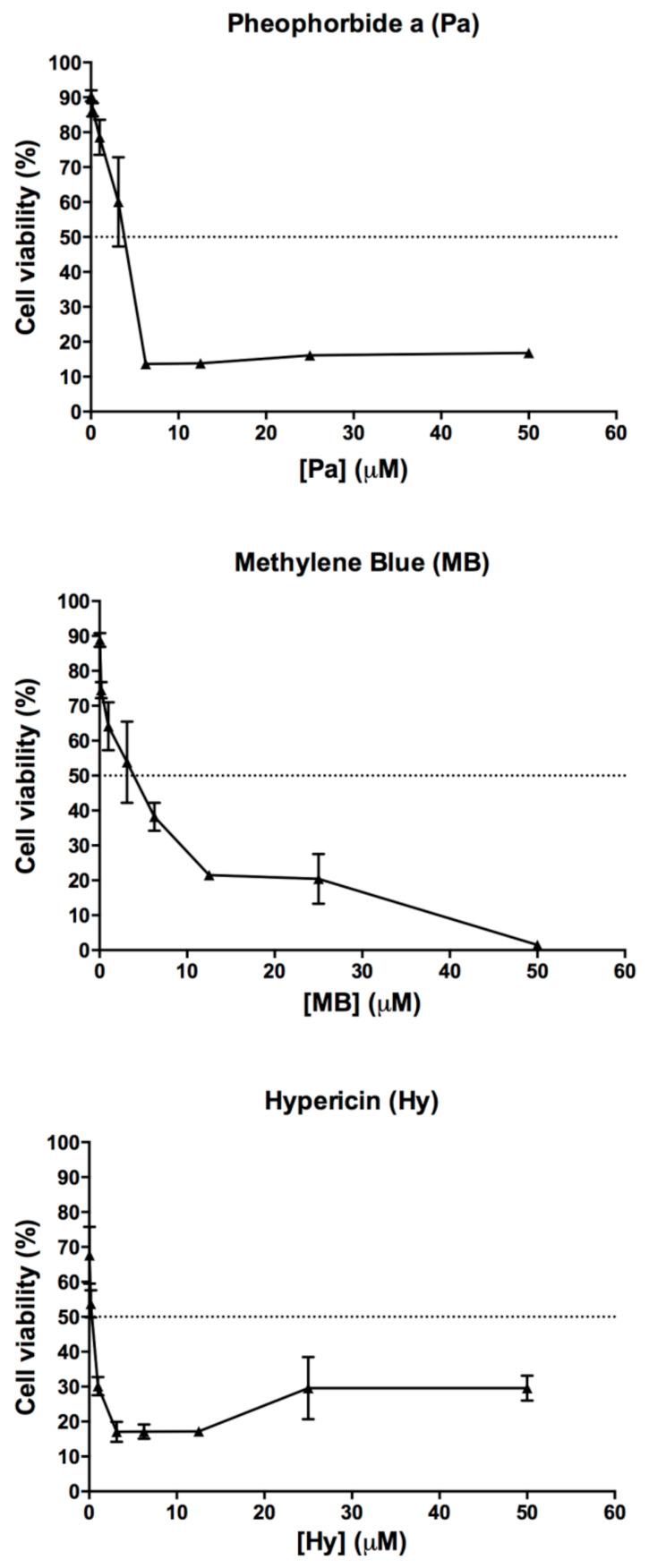
Cytotoxicity of Pa, MB and Hy, with light irradiation, on human neutrophils isolated from buffy coat with. Data are mean ± SEM (*n* = 3).

**Table 1 pharmaceutics-13-01399-t001:** Electronic absorption and basic photophysical data for 3 photosensitizers used.

Compound	λ_max_/nm	λ_em_/nm	Φ_F_ ^a^	Φ_Δ_ ^b^	Ref
Methylene Blue	664 (monomer in aqueous medium)	709	0.04	0.5	[38,39]
Hypericin	598 (DMSO)	651	0.2	0.73	[40,41]
Pheophobide a	667 (DMSO)	677	0.26	0.62	[42,43]

^a^ Fluorescence quantum yield; ^b^ Singlet oxygen quantum yield.

**Table 2 pharmaceutics-13-01399-t002:** The Minimal Bactericidal Concentrations (MBCs) of Pa, Hy and MB against sixteen MRSA strains.

MRSA Type	Strain	MBC Values											
		Hy-PDT		Hy Dark Control		Pa-PDT		Pa Dark Control		MB-PDT		MB Dark Control	
		µM	µg/mL	µM	µg/mL	µM	µg/mL	µM	µg/mL	µM	µg/mL	µM	µg/mL
ATCC	43300	>35	>16	>35	>16	>300	>128	>300	>128	160	32	>160	>32
ATCC	BAA 42	>35	>16	>35	>16	>300	>128	>300	>128	>160	>32	>160	>32
ATCC	BAA 43	>35	>16	>35	>16	>300	>128	>300	>128	80	32	>160	>32
ATCC	BAA 44	>35	>16	>35	>16	>300	>128	>300	>128	>160	>32	>160	>32
Mutant	APH2AAC 6	>35	>16	>35	>16	>300	>128	>300	>128	>160	>32	>160	>32
Mutant	RN4220 /pUL5054	0.625	0.5	5	4	6.25	4	50	32	120	>32	>160	>32
CA ^a^	W44	0.625	0.5	3.125	2.5	12.5	8	75	48	140	>32	>160	>32
CA	W45	10	8	>35	>16	>300	>128	>300	>128	>160	>32	>160	>32
CA	W46	1.25	1	7.5	6	6.25	4	31.25	20	>160	>32	>160	>32
CA	W47	5	4	>35	>16	3.125	2	18.75	12	>160	>32	>160	>32
CA	W48	1.25	1	5	4	>300	>128	>300	>128	140	>32	>160	>32
HA ^b^	W231	>35	>16	>35	>16	>300	>128	>300	>128	>160	>32	>160	>32
HA	W232	2.5	2	15	12	>300	>128	>300	>128	>160	>32	>160	>32
HA	W233	>35	>16	>35	>16	>300	>128	>300	>128	120	>32	>160	>32
HA	W234	>35	>16	>35	>16	>300	>128	>300	>128	80	32	>160	>32
HA	W235	>35	>16	>35	>16	>300	>128	>300	>128	>160	>32	>160	>32

^a^ CA: community associated; ^b^ HA: hospital associated.

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
