# Peer review of "Hypericin and Pheophorbide a Mediated Photodynamic Therapy Fighting MRSA Wound Infections: A Translational Study from In Vitro to In Vivo"

_pharmaceutics, 2021, doi:10.3390/pharmaceutics13091399_

Round 1

Reviewer 1 Report

Overall, the article presents interesting results, despite not showing any efficacy for methylene blue, in bacterial load of wounds, which is already well known. Here are some considerations:

  • At line 68: I suggest making a more detailed description including wavelength, application body region and if possible total radiant energy [J] used.
  • For pdt, the photosensitizer needs to be excited at a certain wavelength. Are there studies showing the excitation of hypericium 590nm and pheophorbide by 610nm? if yes, include in the introduction. If not, how do you know the hypericim and pheophorbide are excited at this wavelength?
  • At figure 3. Just application of light is not PDT. Always PDT is with photosensitizer.
  • Is there any work showing the distribution of Hy and Pa in the skin after intravenous administration? in the dose used.
  • At discussion: please, support your results with more literature citation, as weel as MB efficacy that is well know.
  • what was the concentration of MB used?

Reviewer 2 Report

This manuscript deals with the in vitro and in in vivo screening of Hypericin (Hy) and Pheophobide a (Pa) mediated PDT against a panel of clinically relevant MRSA. Though the authors have done a plenty of work, they failed to include the necessary details in the manuscript. There are some concerns with the manuscript which requires explanation from the authors. The rationale for including Phephobide A should be included in the introduction. The preparation/composition of injection and topical formulation of Hy and Pa used for in vivo studies should be included. It is not clear how the dose and treatment cycles were decided for in vivo studies. The authors should mention this in the manuscript. The authors should explain more about figures in the results section. Most figures are explained with a single sentence. In the discussion the authors stated that “there is complete elimination of MRSA by Fucidin cream treatment, but wound healing as reflected by large open wound areas is worst”. Any possible reason for such behavior?

Reviewer 3 Report

In this work, Chan and co-workers studied the use of hypericin and pheophorbide a as photosensitizers toward antimicrobial photodynamic therapy (aPDT). Namely, they studied the light-mediated antimicrobial activity of these compounds toward a panel of clinically relevant methicillin-resistant staphylococcus aureus (MRSA), and evaluated also the selected compounds in full-thick MRSA wound infection in vivo models. Finally, they assessed also the interaction of aPDT with the immune system via cytotoxicity toward neutrophils.

This topic of research is quite relevant, given that aPDT could be quite important for treatment of antibiotic-resistant bacteria. These studied compounds provided also efficiency gains when in comparison with a reference photosensitizer (methylene blue). However, the authors must address some questions/comments, before recommendation for acceptance can be given:

-The authors should be more clear on the previous use of both hypericin and pheophorbide a in both anticancer PDT and in aPDT, in the Introduction section.

-The photochemical/photophysical characterization of both hypericin and pheophorbide a is significantly incomplete. More specifically, the manuscript lacks information about the absorption, excitation and fluorescence spectra of these compounds. Important parameters such as fluorescence quantum yields and singlet oxygen sensitization yields are also missing. This information can be measured by the authors, or cited if already indicated in the literature. Either way, all this data is necessary.

- On the first paragraph of section 3.1, the authors state that PDT toxicities are higher than dark toxicities. However, this also means that is dark toxicity. The authors should present these values.

-Line 268 appears to be strange.

-The authors indicated that Hy-PDT and Pa-PDT showed better efficacy than MB-PDT in wound healing effects. However, none of these approaches appeared to be particular effective in vivo. Thus, it would be useful if a positive control (known antibacterial agents) was also included in the study, to see if aPDT in general has any particular advantage over other antibacterial agents.

-The studied systems (both in the presence and absence of light) showed relevant toxicity toward neutrophils (except maybe Hy, in the absence of light). Does this not indicate that these compounds are toxic toward human cells, and so, ineffective toward in vivo applications?

Reviewer 4 Report

Undoubtedly, the topic of the manuscript is very important due to the problem of antibiotic resistance. aPDT has proven to be effective against MRSA infections, which is documented by a large number of original articles and review papers. Among these countless published papers, a very small percentage of them address the problem of testing on in vivo models of bacterial infections. And this is extremely important if we aim, and we do aim, to translate preclinical research to clinical.

 Therefore, in this aspect, the peer-reviewed article could have contributed something new to the field. Unfortunately, this is not the case for several reasons.

  1. The photosensitizers used in these studies are commercial compounds that have been tested by many scientific groups; the authors could potentially improve their outcome by using some innovative formulations, which in fact would be a scope of the journal.
  2. The studies were inappropriately designed, particularly with regard to the proper choice of PS doses, light doses, and treatment protocols in general.
  3. The studies performed should be presented in a similar manner to that available in the literature. A further problem arises here, though, as in my opinion the selection of the current literature is at least unfortunate.

Therefore, in its current form, the manuscript is not suitable for publication.

Below I present the most important errors and shortcomings that need to be changed/corrected.

  • Please in Materials and Methods, please present all the experimental details, including irradiation area and exactly calculated light doses delivered to the wound.
  • Please introduce Kaplan–Meier wound healing curves of MRSA infected mouse abrasion wounds without treatment (neither PS nor light was applied) and treated with aPDT.
  • Introduce the FIGURE that help us to see the monitoring of aPDT of infected skin abrasions during light delivery.

I have to admit that to the best to my knowledge in in vivo settings light doses even up to 200-240 J/cm-1 are required!

  • Also results of in vitro experiments should be presented in the light dose depended manner.
  • In the Discussion section it is written (line 339) that 570 nm is a NIR range. It is not. NIR part of light starts around 700, even 730 nm.
  • In Fig. 10 and 11 the authors assigned the results from MTT test as cell viability. It is not correct. MTT does not inform us on cell viability but rather on mitochondrial activity.
  • In Fig. 11 the authors demonstrated that studied Pss are active against human neutrophils. I should say, that it is rather disadvantage. The best photosensitizers described in the literature are those generating significant amount of ROS to kill microbes at the drug dose and light dose that not affecting normal human cells.

Round 2

Reviewer 1 Report

Dear, I felt contemplated with the answers. I agree with the acceptance of the publication.

Reviewer 3 Report

The authors have addressed my comments, and so, acceptance is recommended.